# Functional Recovery after Surgery for Lumbar Spinal Stenosis in Patients with Hypertension

**DOI:** 10.3390/healthcare8040503

**Published:** 2020-11-20

**Authors:** Sanjesh C. Roop, Michele C. Battié, Gian S. Jhangri, Richard W. Hu, C. Allyson Jones

**Affiliations:** 1School of Public Health, University of Alberta, Edmonton, AB T6G 1C9, Canada; sanjeshroop@gmail.com (S.C.R.); gian.jhangri@ualberta.ca (G.S.J.); 2School of Physical Therapy, Faculty of Health Sciences, University of Western Ontario, London, ON N6G 1H1, Canada; mbattie@uwo.ca; 3Section of Orthopaedics, Department of Surgery, University of Calgary, Calgary, AB T2N 2T9, Canada; Richard.Hu@albertahealthservices.ca

**Keywords:** lumbar spinal stenosis, hypertension, surgery, function, recovery

## Abstract

Hypertension is a prevalent condition that is associated with lower health status in patients with lumbar spinal stenosis. The study determined whether hypertension is a prognostic factor associated with functional recovery after spine surgery for lumbar spinal stenosis. This was a secondary analysis of the Alberta Lumbar Spinal Stenosis Study in which patients were identified as participants at the time of lumbosacral magnetic resonance imaging or computed tomography in Calgary, Alberta, Canada. Multivariable linear regression analyses were performed to examine hypertension as a prognostic factor of functional recovery after surgery (Oswestry Disability Index, the Swiss Spinal Stenosis (SSS)-Symptom Severity and SSS-Physical Function scales). Of the 97 surgical participants, 49 who were hypertensive were older (76.8, SD 11.4 years) than the 48 non-hypertensive participants (66.7, SD 12.4 years) (*p* < 0.001). No significant associations between hypertension and post-operative function in any of the three multivariable models were seen. The Oswestry Disability Index mean score improved after surgery (effect size: 1.73; 95%CI: 1.39, 2.06), with no differences seen between those with and without hypertension (*p* = 0.699). Large changes were seen after surgery for the SSS-Symptom Severity (effect size: 1.0, 95%CI 0.7, 1.3) and SSS-Physical Function (effect size: 0.9, 95%CI 0.6, 1.2) scales. Hypertension alone does not negatively impact functional recovery following surgery.

## 1. Introduction

Lumbar spinal stenosis is one of the most commonly diagnosed spinal conditions and the leading indication for spine surgery in adults aged 65 years and older [1,2,3,4]. Diagnosis and treatment of lumbar spinal stenosis is complex, and the risks related to spine surgery are a serious consideration for older adults [5]. When conservative management is no longer of benefit, spine surgery for lumbar spinal stenosis is an option that can provide pain relief and improve function; however, at least 30% of patients will report back pain over the long term post-operatively [4,5,6]. Several surgical options exist for lumbar spinal stenosis with the primary goal to decompress the affected neural structures. Not only do comorbidities have a negative influence on complications and mortality after spine surgery [7], but they also have deleterious effects on pain and functional recovery [1,6,8,9]. Using comorbidity indices, cardiovascular comorbidity is reported to have a consistent deleterious effect on post-operative pain and function after surgery for lumbar spinal stenosis [9]. Within a community-based cohort of lumbar spinal stenosis, hypertension was also associated with lower health status [10].

Hypertension (>140/90 mmHg) is a prevalent condition with 23% reported in the Canadian general adult population [11]. Patients diagnosed with lumbar spinal stenosis have more chronic conditions, in particular, a higher prevalence of hypertension than hospital controls [12] and the general population [10,13]. In a community-based cohort of participants with lumbar spinal stenosis, the odds of having hypertension were 1.70 (95% CI: 1.27, 2.28) times greater than the general population of the same age [10]. After adjusting for age and comorbidities, the health status of lumbar spinal stenosis patients was approximately four times the clinical important difference lower than the general population [10]. Systemic diseases including hypertension have been associated with the pathophysiology of spinal stenosis, in that impaired blood flow related to hypertension has been postulated to facilitate degenerative changes in the spine [14].

The relationship among the cardiovascular system, pain and function is complex. Although work has focused on nociceptive response and its effect on blood pressure in experimental studies [15,16], the effect of hypertension on pain perception is less clear especially in light of comorbidities such as chronic pain [16] and spinal conditions [14].

A community-based cohort at the time of diagnostic imaging was assembled and followed over time to identify participants who proceeded to surgery. The primary aim was to determine whether hypertension is a prognostic factor associated with functional recovery after spine surgery for lumbar spinal stenosis. Within this surgical subset, we examined the prognostic value of hypertension on functional recovery after surgery for lumbar spinal stenosis. Based on evidence that reported hypertension was associated with lower health status in a community-based cohort with lumbar spinal stenosis [10], we hypothesized that hypertension prior to surgery would be a prevalent comorbidity in this surgical cohort and would also be a significant prognostic factor associated with poor functional recovery after spine surgery for lumbar spinal stenosis. Providing a better understanding of the impact of chronic conditions on functional recovery from spine surgery will help patients and health professionals make informed decisions as to whether to proceed with surgery and to predict successful recovery processes.

## 2. Materials and Methods

### 2.1. Study Design and Participants

This was a secondary analysis of the Alberta Lumbar Spinal Stenosis Study, a prospective cohort study of patients who were identified as study candidates at the time of lumbosacral magnetic resonance imaging or computed tomography. Patients were referred to one of four imaging centers in Calgary, Alberta, Canada, by either general practitioners or specialists to investigate possible lumbar spinal stenosis. To be considered for inclusion to the primary study, a clinical radiological report indicating central or combination lumbar spinal stenosis at one or more levels was required. Anatomic lumbar spinal stenosis was defined as narrowing of the central spinal canal, lateral recesses or neural foramen because of encroachment by surrounding bone and soft tissue [17]. Other inclusion criteria were: aged 40 years or older given the increasing surgical rates in older adults [7]; English speaking; referral to imaging with back and/or leg pain; no spinal malignancies, infections, inflammatory conditions or fractures; and no active cancer for which metastases were suspected. Of the patients referred to the 4 imaging centers who volunteered to participate, 1047 participants were 40 years or older and had some aspect of anatomic stenosis noted in their clinical radiology report. We were able to contact 710 participants by telephone to complete study questionnaires and identified 245 who had received the clinical diagnosis of lumbar spinal stenosis [10].

Eligibility for this analysis required that participants had (1) lumbar spinal stenosis-related surgery during the study period, and (2) a baseline interview prior to surgery and a follow-up interview within 2 years after surgery. Informed consent was obtained from all individual participants included in the study. Participants were also asked to sign a consent to access their administrative hospital data from the provincial Ministry of Health, Alberta Health. The study was approved by the Research Health Ethics Boards at the University of Alberta (Pro00003240) and the University of Calgary (ID17024).

Participants were interviewed upon entry to the study and annually afterward. For each interview, participants were asked whether they had undergone surgery for lumbar spinal stenosis. They were also asked questions concerning socio-demographic characteristics (age, gender, education, marital status, employment status), signs and symptoms (symptom severity and duration, pain, physical function, claudication, walking ability), depressive symptomology (Centre for Epidemiologic Studies Depression (CES-D) Scale) [18], specific comorbidities and medications. They also completed two spine-specific functional measures, the Oswestry Disability Index (ODI) [19,20] and the Swiss Spinal Stenosis (SSS) questionnaire [21].

Comorbid conditions were identified from a list of predefined conditions obtained from the Charlson Comorbidity Index [22] and the Canadian National Population Health Survey [23]. The specific comorbidities were derived from self-report and/or hospital admission conditions listed in the Alberta Health data. At the baseline interview, participants identified comorbidities they were currently experiencing.

Surgical cases were ascertained using data from the interviews and administrative health data. The Alberta Health data included: Inpatient Discharge Abstract Database and Physician Claims files. All administrative health databases are maintained by Alberta Health for the Health Care Insurance Plan. Lumbar spinal stenosis-related surgery was identified from surgical procedure codes (1.SC.74 “spinal vertebrae, fixation”; 1.SC.75 “spinal vertebrae, fusion”; 1.SC.89 “excision total, spinal vertebrae”; 1.SE.53 “implantation of internal device, intervertebral disc”; 1.SC.80 “repair, spinal vertebrae”; 1.SE.87 “excision, partial, intervertebral disc”; and 1.SE.89 “excision total, intervertebral disc”). To confirm lumbar spinal stenosis surgical cases, ICD9/10-CM diagnostic codes were used for each hospital admission.

Participants with hypertension were identified by self-report and International Classification of Diseases, Ninth Revision (ICD9) and Tenth Revisions (ICD10-CM) codes for hypertension reported in the administrative health data. During the interview, participants were asked if they currently have or are being treated for hypertension. Those participants with ICD9/10-CM codes for hypertension at hospital admission (ICD9: 401.0 “malignant essential hypertension”; 401.1 “benign essential hypertension”; and 401.9 “unspecified essential hypertension”) were also considered hypertensive cases.

Within this community-based cohort, 122 participants were identified as having back surgery after the imaging date. Nineteen (15.6%) of the 122 participants did not have a pre-operative interview, and another 6 (4.9%) participants were excluded because the post-operative interview was either missing or occurred more than 2 years after surgery. The remaining 97 participants were classified as the surgical cohort, of which 46 reported having hypertension prior to surgery. An additional 3 participants were identified via administrative records as having hypertension, resulting in a total of 49 (50.5%) hypertensive participants.

### 2.2. Measurements

The classic symptoms of LSS include numbness, cramping in the legs and pain which are aggravated by walking and standing [5]. These symptoms have ramifications on functional activities of daily living.

The ODI [19,20], which was designated as the primary outcome, is a 10-item measure that measures function affected by low back pain. A single summary score is generated with scores between 0 and 20% indicating minimal disability, 20 and 40% moderate disability, 40 and 60% severe disability and scores above 60% representing severely disabling pain [24]. A minimally clinically important difference of 12.8 units has been reported for the ODI in a lumbar spinal stenosis surgical patient population [25].

The secondary outcome examined was the SSS questionnaire, a disease-specific measure for lumbar spinal stenosis [21]. For the purposes of this study, we looked at 2 of the 3 SSS subscales. The 7-item Symptom Severity (SSS-SS) subscale uses a 5-point Likert scale to capture pain and the neuro-ischemic symptoms seen with LSS [3]. These features are not otherwise captured by a generic low back pain outcome measure. The other subscale was the 5-item Physical Function (SSS-PF) subscale which uses a 4-point Likert scale. Each scale is expressed as a mean, with greater scores indicative of worse disability.

### 2.3. Statistical Analysis

Since post-operative outcomes were dependent upon recovery time, post-operative interviews were statistically adjusted within the prognostic model as one of two classifications. Participants whose follow-up interview occurred within 6 months of surgery were classified as sub-acute, whereas participants who were interviewed from 6 months to 2 years were classified as long-term. Summary statistics and univariate analyses were performed. Standardized effect sizes were calculated for the ODI, SSS-SS and SSS-PF by the difference between the pre- and post-operative scores divided by the baseline standard deviation [26]. A positive value for the effect size indicated improvement over the time interval, whereas a negative value indicated deterioration.

Multivariable linear regression analyses were performed to examine hypertension as a prognostic factor of functional recovery for spine surgery as indicated by the ODI, SSS-SS and SSS-PF. A risk factor modeling strategy was used by entering variables separately into multivariable linear regression models [27]. The hypertension variable and possible confounding variables were examined separately at a univariate stage of initial modeling development. Variables were selected for the parsimonious model because of potential confounding effects and included age, gender, follow-up time, number of comorbid conditions and depression as deemed biologically important variables. To be considered a confounding variable, the hypertension coefficient had to change at least 15% with the addition of each variable to the model. This approach permitted the inclusivity of variables which were most likely to have a confounding effect. A *p* < 0.05 was considered for statistical significance for the final model. All analyses were performed using SPSS, version 21 (SPSS, Inc., Chicago, IL, USA).

## 3. Results

The mean age of this surgical cohort was 71.8 (SD 12.9) years, with 52 (53.6%) being female (Table 1). The 49 patients with hypertension were older (76.8, SD 11.4 years) compared to the 48 participants without hypertension (66.7, SD 12.4 years) (*p* < 0.001). Excluding hypertension as a comorbidity, participants without hypertension had significantly fewer (1.7, SD 1.2) comorbidities than the hypertensive group (2.9, SD 2.2) (*p* = 0.001). The three most prevalent conditions for the hypertensive group were heart disease (*n* = 18; 36.7%), urinary incontinence (*n* = 17; 34.7%) and diabetes mellitus (*n* = 12; 25.0%). The most common conditions for the non-hypertensive group were depression (*n* = 20; 41.7%), heart disease (*n* = 11; 22.9%) and urinary incontinence (*n* = 9; 18.8%). The mean number of medications reported was 3.2 (SD 1.4) and did not differ between the two hypertensive groups (*p* = 0.200). The median length of hospital stay was 5.0 (interquartile range 3.0–8.0) days. Decompressive laminectomy accounted for nearly half of the surgical procedures (*n* = 25, 49.0%). The second most common surgical procedure was spinal fusion (*n* = 23, 45.0%). The median pre-operative time from interview to surgery was 4.1 (interquartile range 1.0–7.3) months, whereas the median post-operative follow-up time was 7.3 (interquartile range 5.4–11.7) months. No significant differences in times were seen between the two groups (*p* > 0.05).

The pre-operative ODI mean score for this cohort was 59.0 (SD 17.0), which represented severe disability, with no differences seen between the two blood pressure groups (*p* = 0.976) (Table 2). Although the ODI mean score improved after surgery (29.4, SD 17.3), no between-group differences were seen (*p* = 0.699). A similar pattern was seen with the SSS-SS and SSS-PF subscales. Large changes were seen after surgery for the SSS-SS (effect size: 1.0, 95%CI 0.7, 1.3) and SSS-PF (effect size: 0.9, 95%CI 0.6, 1.2) scales, with no differences seen between the hypertensive groups (Table 2).

The influence of hypertension was examined in the three multivariable models fitted for post-operative function (ODI, SSS-SS, SSS-PF scores) adjusting for age, gender, follow-up time, pre-operative functional score, number of chronic conditions and depression. There were no statistically significant associations between hypertension and the three post-operative function measures either in univariate or multivariable models (Table 3).

## 4. Discussion

While half of this surgical cohort was hypertensive, pre-operative hypertension did not have a deleterious effect on recovery up to two years after surgery for lumbar spinal stenosis. Findings suggest that patients undergoing surgery for lumbar spinal stenosis can expect large functional gains, which is consistent with prior studies [28,29,30]. Within this universal healthcare system, a relatively small proportion of patients received surgery within the first few years of referral for imaging. While there has been little investigation of the relationship between hypertension and lumbar spinal stenosis, even less work has evaluated the impact of hypertension on functional outcomes after surgery for lumbar spinal stenosis [31]. Hypertension was reported as a risk factor for adverse events with complex elective lumbar fusion [32]. Conversely, self-reported hypertension was not statistically significantly associated with condition-specific or generic health outcomes at one-year follow up in 1329 patients who received spine surgery for lumbar spinal stenosis [31].

Based on our findings, hypertension does not appear to affect recovery after surgery for lumbar spinal stenosis, yet others have reported that comorbidities of lumbar spinal stenosis are associated with health-related quality of life in patients with lumbar spinal stenosis [1,10,33]. Battié and colleagues examined the overall health of participants with lumbar spinal stenosis and associated comorbidities. Using the Health Utilities Index Mark 3 (HUI3) [34], a generic health measure, they reported that lumbar spinal stenosis comorbidities, including hypertension, resulted in lower HUI3 scores after adjusting for age and gender [10]. The discrepancy of hypertension with lumbar spinal stenosis may be due to a few factors. We used spine-specific outcome measures (ODI, SSS-SS and SSS-PF) to examine hypertension as a prognostic factor, whereas Battié and colleagues used a generic health measure. A generic health measure such as the HUI3 evaluates overall health and is more likely responsive to the effect of other conditions than a spine-specific measure (i.e., ODI). They also looked at baseline associations, cross-sectionally, in a lumbar spinal stenosis cohort regardless of future surgical status.

Regardless of blood pressure status, large gains in function and reduction in symptoms were seen up to two years after surgery for lumbar spinal stenosis, with a large effect size reported. When comparing mean pre- and post-operative ODI scores, the surgical cohort showed a clinically meaningful difference for both blood pressure groups. It should be acknowledged that no clinically meaningful differences for the ODI have been defined in surgical patients with lumbar spinal stenosis; however, a 12.8 point difference in the ODI has been defined as a clinically meaningful difference for non-surgical lumbar spinal stenosis patients [25].

While a strength of this study was the recruitment of patients with lumbar spinal stenosis from the community rather than spine centers or specific surgical practices, limitations center on the small surgical sample. Since participants were identified at the time of diagnosis, participants were followed over the course of their chronic condition, lumbar spinal stenosis, to surgery, suggesting a small proportion go on for surgery. It should also be recognized that only those patients who were medically stable were selected for elective lumbar spinal stenosis surgery.

Although earlier work suggested that people with hypertension and lumbar spinal stenosis have lower overall health-related quality of life than those without hypertension, hypertension does not appear to have a deleterious effect on functional recovery after lumbar spinal stenosis-related surgery. All the same, confirmatory studies are warranted with distinct lumbar spinal stenosis cohorts to provide a fuller understanding of the effects of hypertension. These findings do suggest that patients undergoing surgery for lumbar spinal stenosis can expect large gains in functional recovery and symptom reduction up to two years after surgery. For patients with hypertension who are concerned about undergoing lumbar spinal stenosis-related surgery, hypertension alone does not appear to negatively impact functional recovery.

## Figures and Tables

**Table 1 healthcare-08-00503-t001:** Baseline characteristics of lumbar spinal stenosis participants with and without hypertension.

Characteristics	Hypertensive (*n* = 49)	Non-Hypertensive (*n* = 48)
Age, years, mean ± SD	76.8 ± 11.4	66.7 ± 12.4
Gender, female, *n* (%)	28 (57.1)	24 (50.0)
Education, completed high school, *n* (%)	41 (83.7)	46 (95.8)
Marital Status		
Married/common law, *n* (%)	34 (69.4)	34 (70.8)
Living situation, *n* (%)		
Live alone	9 (18.4)	8 (16.7)
Living with others	40 (81.6)	39 (81.3)
Number of comorbidities, mean ± SD	2.5 ± 1.9	1.5 ± 1.1
Depressive symptomology, *n* (%)		
* CES-D 19+ score	26 (53.1)	21 (43.8)
Comorbidity, *n* (%)		
Heart disease	18 (36.7)	10 (20.8)
Incontinence	16 (32.7)	8 (16.7)
Diabetes	12 (24.5)	4 (8.3)
Thyroid disorder	10 (20.4)	2 (4.2)
Cancer	8 (16.3)	4 (8.3)
Bowel disorder	7 (14.2)	3 (6.3)
Obesity	6 (12.2)	4 (8.3)
Respiratory disorder	6 (12.2)	3 (6.3)
Arthritis	3 (6.1)	5 (10.4)
Stroke	2 (4.1)	1 (2.1)
Peripheral vascular disease	2 (4.1)	0

* CES-D, Centre for Epidemiologic Studies Depression Scale; cut-off score 19+.

**Table 2 healthcare-08-00503-t002:** Health measure summary scores and effect size.

Health Outcome	Pre-Operative Score Mean (SD)			Post-Operative Score Mean (SD)			
	Hypertension (*n* = 49)	Without Hypertension (*n* = 48)	*p*-Value	Hypertension * (*n* = 49)	Without Hypertension † (*n* = 48)	*p*-Value	Effect Size ‡ (95% CI)
Oswestry Disability Index	58.9 (18.7)	59.0 (15.1)	0.98	30.1 (17.7)	28.7 (17.1)	0.70	1.73 (1.39, 2.06)
Swiss Spinal Stenosis, Symptom Severity	2.2 (0.72)	2.1 (0.59)	0.91	1.4 (0.83)	1.4 (0.79)	0.91	1.00 (0.70, 1.30)
Swiss Spinal Stenosis, Physical Function	2.5 (0.61)	2.4 (0.64)	0.15	1.9 (0.74)	1.8 (0.64)	0.29	0.92 (0.62, 1.22)

* Two patients have missing values for Oswestry Disability Index (ODI), 3 for Swiss Spinal Stenosis-Symptom Severity (SSS-SS) and 4 for SSS-Physical Function (PF). † One patient has a missing value for SSS-PF. ‡ Effect size = (baseline score—post-operative score)/standard deviation of baseline score.

**Table 3 healthcare-08-00503-t003:** The relationship between hypertension and post-operative health outcomes after surgery for lumbar spinal stenosis.

Outcome	Coefficient (95%CI)	Adjusted * Coefficient (95% CI)
Oswestry Disability Index		
Post-operative score	1.39 (−5.72, 8.49)	1.32 (−5.64, 8.28)
Swiss Spinal Stenosis		
Symptom Severity	0.02 (−0.31, 0.35)	0.03 (−0.33, 0.39)
Physical Function	0.15 (−0.13, 0.44)	0.17 (−0.15, 0.49)

* Adjusting for age, gender (female), follow-up time (<6 months), pre-operative score, number of comorbidities and depression (Centre for Epidemiologic Studies Depression Scale; cut-off score 19+). CI = confidence interval.

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
