# Peer review of "Functional Recovery after Surgery for Lumbar Spinal Stenosis in Patients with Hypertension"

_healthcare, 2020, doi:10.3390/healthcare8040503_

Round 1
Reviewer 1 Report
The present study by Roop et al. sought to determine whether hypertension predicts functional recovery after spine surgery for lumbar spinal stenosis. The authors report no significant associations between hypertension and post-operative function. The study addresses an interesting question that could potentially inform clinical decisions. There are also several concerns that should be addressed.
1. The stated hypothesis should be clearer. How would hypertension be associated with functional recovery? Lower functional recovery? This would help the reader follow the rationale for the study. Also, the introduction is missing the physiological connection between hypertension and post-operative results. How would hypertension influence functional recovery on physiological level? The body of literature regarding hypertension and post-operative pain symptoms could be referenced with mention of relevant physiological mechanisms (e.g., baroreceptors, endogenous opioids). For example, Bruehl & Chung. Neuroscience and Biobehavioral Reviews. 28:395-414. 2004.
2. Although functional recovery is important, it seems that improvement in symptoms (e.g., pain, numbness, etc.) would be equally as important. It is indicated that symptom severity and duration and pain were assessed (line 96-97). These data should be included and analyzed like functional recovery. This could potentially add significant weight to the manuscript because there is a physiological rationale for a blood pressure-pain interaction.
3. Data regarding medications, including pain-related medications, are missing from the manuscript. What were the relevant medications and how did medications impact the findings? Were there differences between groups (hypertensive vs. non-hypertensive) in medications?
4. Consider including a rationale for using 40 years of age or older as inclusion criteria.
Author Response
healthcare-974396
Functional recovery after surgery for lumbar spinal stenosis in patients with hypertension
We would like to thank the reviewers for their thoughtful comments. We have responded to their concerns below and included the line number(s) with the revisions. If it is easier, we can also submit a version with the revisions (highlighted) in the text.
reviewer 1
- The stated hypothesis should be clearer. How would hypertension be associated with functional recovery? Lower functional recovery? This would help the reader follow the rationale for the study. Also, the introduction is missing the physiological connection between hypertension and post-operative results. How would hypertension influence functional recovery on physiological level? The body of literature regarding hypertension and post-operative pain symptoms could be referenced with mention of relevant physiological mechanisms (e.g., baroreceptors, endogenous opioids). For example, Bruehl & Chung. Neuroscience and Biobehavioral Reviews. 28:395-414. 2004.
>> L 62-67. We revised the hypothesis to provide more specifics about the functional recovery and pre-operative hypertension. Thank you for the reference.
L 55-58. We included a section to discuss the relationship of pain and hypertension in the introduction.
- Although functional recovery is important, it seems that improvement in symptoms (e.g., pain, numbness, etc.) would be equally as important. It is indicated that symptom severity and duration and pain were assessed (line 96-97). These data should be included and analyzed like functional recovery. This could potentially add significant weight to the manuscript because there is a physiological rationale for a blood pressure-pain interaction.
>> L 132-134; L142-145. The SSS-SS captures pain and the neuro ischemic symptoms seen with LSS. Comer and colleagues when evaluating the construct validity of the SSS reported that the symptom severity scale measures pain in the back and buttocks and that it contributes to the functional problems. These features are not otherwise captured by a generic low back pain outcome measure. We did not include a generic pain measure as this would capture pain not only in the back.
We added further detail in the description of the measure to acknowledge the measurement of pain within the SSS-SS.
- Data regarding medications, including pain-related medications, are missing from the manuscript. What were the relevant medications and how did medications impact the findings? Were there differences between groups (hypertensive vs. non-hypertensive) in medications?
>> L 176-178. Medications were classified into categories included anti-depressants; muscle relaxants; narcotics; analgesics; sedatives and steroid medication. No statistical differences were seen between the 2 groups. We included description of this in the results.
- Consider including a rationale for using 40 years of age or older as inclusion criteria.
>> L81. Symptomatic degenerative lumbar spinal stenosis is primarily seen in older patients. Although the prevalence has not been established, increasing surgical rates in older adults are reported, 65 yrs.1 A 40 yr age cut-off was used within the primary study given that the surgery is typically seen in later years.2 A rationale was provided in the inclusion criteria.
Reviewer 2 Report
The manuscript reports the data on the association between hypertension and post-surgery spinal stenosis outcomes. No impact of hypertension on functional recovery after surgery was found. Despite the negative findings, the study is scientifically sound and no major concerns were noted.
There are minor concerns in the style such as missing spaces after a period or comma, missing comas, large spaces between paragraphs that should be all edited.
In addition, The manuscript is text heavy and repeats information, e.g. findings in results are stated again in table 1-3. On the other hand, in the introduction or discussion, the authors can expand beyond the reference 12 on why hypertension is causing degenerative changes in the spine.
Author Response
healthcare-974396
Functional recovery after surgery for lumbar spinal stenosis in patients with hypertension
We would like to thank the reviewers for their thoughtful comments. We have responded to their concerns below and included the line number(s) with the revisions. If it is easier, we can also submit a version with the revisions (highlighted) in the text.
REVIEWER 2
The manuscript reports the data on the association between hypertension and post-surgery spinal stenosis outcomes. No impact of hypertension on functional recovery after surgery was found. Despite the negative findings, the study is scientifically sound and no major concerns were noted.
There are minor concerns in the style such as missing spaces after a period or comma, missing comas, large spaces between paragraphs that should be all edited.
>> Thank you. We have reviewed and tried to correct all editing errors.
In addition, The manuscript is text heavy and repeats information, e.g. findings in results are stated again in table 1-3. On the other hand, in the introduction or discussion, the authors can expand beyond the reference 12 on why hypertension is causing degenerative changes in the spine.
>> L 55-58; L 211-214. We have included additional explanation in the Discussion section.